# The Superiority of *Bacillus megaterium* over *Escherichia coli* as a Recombinant Bacterial Host for Hyaluronic Acid Production

**DOI:** 10.3390/microorganisms10122347

**Published:** 2022-11-28

**Authors:** HebaT’Allah Nasser, Bernhard J. Eikmanns, Mahmoud M. Tolba, Mohamed El-Azizi, Khaled Abou-Aisha

**Affiliations:** 1Department of Microbiology, Immunology, and Biotechnology, Faculty of Pharmacy and Biotechnology, German University in Cairo, Cairo 11435, Egypt; 2Institute of Microbiology and Biotechnology, Ulm University, 89081 Ulm, Germany; 3Pharmaceutical Division, Ministry of Health and Population, Faiyum City 63723, Egypt

**Keywords:** Hyaluronan, *Bacillus megaterium*, *Escherichia coli*, *Streptococcus equi* subsp. *zooepidemicus*, *hasABCDE* genes

## Abstract

(1) Background: Hyaluronic acid (HA) is a polyanionic mucopolysaccharide extensively used in biomedical and cosmetic industries due to its unique rheological properties. Recombinant HA production using other microbial platforms has received increasing interest to avoid potential toxin contamination associated with its production by streptococcal fermentation. In this study, the Gram-negative strains *Escherichia coli* (pLysY*/Iq*), *E. coli* Rosetta2, *E. coli* Rosetta (DE3) pLysS, *E. coli* Rosetta2 (DE3), *E. coli* Rosetta gammiB(DE3)pLysS, and the Gram-positive *Bacillus megaterium* (MS941) were investigated as new platforms for the heterologous production of HA. (2) Results: The HA biosynthesis gene *hasA*, cloned from *Streptococcus equi* subsp. *zoopedemicus*, was ligated into plasmid pMM1522 (MoBiTec), resulting in pMM1522 *hasA*, which was introduced into *E. coli* Rosetta-2(DE3) and *B. megaterium* (MS941). The initial HA titer by the two hosts in the LB medium was 5 mg/L and 50 mg/L, respectively. Streptococcal *hasABC* and *hasABCDE* genes were ligated into plasmid pP_T7_ (MoBiTec) and different *E. coli* host strains were then transformed with the resulting plasmids pP_T7_*hasABC* and pP_T7_*hasABCDE*. For *E. coli* Rosetta-gamiB(DE3)pLysS transformed with pP_T7_*hasABC*, HA production was 500 ± 11.4 mg/L in terrific broth (TB) medium. Productivity was slightly higher (585 ± 2.9 mg/L) when the same host was transformed with pP_T7_ carrying the entire HA operon. We also transformed *B. megaterium* (MS941) protoplasts carrying T7-RNAP with pP_T7_*hasABC* and pP_T7_*hasABCDE*. In comparison, the former plasmid resulted in HA titers of 2116.7 ± 44 and 1988.3 ± 19.6 mg/L in LB media supplemented with 5% sucrose and A5 medium + MOPSO, respectively; the latter plasmid boosted the titer final concentration further to reach 2476.7 ± 14.5 mg/L and 2350 ± 28.8 mg/L in the two media, respectively. The molecular mass of representative HA samples ranged from 10^5^ − 10^6^ Daltons (Da), and the polydispersity index (PDI) was <2. Fourier transform infrared spectroscopy (FTIR) spectra of the HA product were identical to those obtained for commercially available standard polymers. Finally, scanning electron microscopic examination revealed the presence of extensive HA capsules in *E. coli* Rosetta-gamiB(DE3)pLysS, while no HA capsules were produced by *B. megaterium*. (3) Conclusions: Our results suggested that Gram-positive bacteria are probably superior host strains for recombinant HA production over their Gram-negative counters. The titers and the molecular weight (MW) of HA produced by *B. megaterium* were significantly higher than those obtained by different *E. coli* host strains used in this study.

## 1. Introduction

HA is a linear mucopolysaccharide with a polyanionic nature and consists of disaccharide repeats of D-glucuronic acid (GlcUA), and N-acetylglucosamine (GlcNAc) joined alternatively by β-1,3 and β-1,4 glycosidic bonds [1,2,3]. Due to its excellent water-binding capacity and viscoelasticity, HA possesses unique physicochemical and rheological properties. Accordingly, HA has been used in many biomedical applications, including chondro-protection, orthopedic surgery, visco-supplementation, anti-adhesion therapy, dermatology, wound-healing in ophthalmology, tissue regeneration in the cardiovascular system, and anti-aging cosmetics [3,4,5,6,7,8]. In addition, HA has been used as a chemotherapeutic drug carrier in conjugated complexes. The HA-Paclitaxel (PTX) antitumor conjugate is an example in which HA provides the dual advantage of accumulation at the tumor site and a receptor-mediated uptake [3,9,10]. HA can also be coupled with fibrin in hydrogel to function as a delivery system for mesenchymal stem-cell injection [11].

Currently, there are two competing methods for the industrial production of HA including the extraction from animal sources, such as bovine eyes and rooster combs, and the large-scale expression in microbial host strains, e.g., in *Streptococcus* [7,12,13]. HA extracted from animal sources suffers from two significant limitations. First, in animal tissues, HA is complexed with proteoglycans and often contaminated with HA-degrading enzymes, making isolating high-purity and large-molecular-size polysaccharides difficult and costly [14]. Second, using animal-derived biomolecules for biopharmaceutical applications is facing growing concerns because of the risk of cross-species viral and other adventitious agent contaminations. The FDA has not approved HA production from animal-derived sources for these reasons. Since the 1980s, recombinant HA microbial production has gradually replaced extraction from animal tissues [2,14]. The commonly used strain for HA production is *Streptococcus equi* subsp*. zooepidemicus* [7,11,15]. In these bacteria, the HA capsule is a virulence factor, which contributes to the pathogenicity, probably providing an unknown immunological response that fails the immune system to recognize the HA capsule as a foreign entity [16,17]. However, HA production by *Streptococcus equi* subsp*. zooepidemicus* is facing some drawbacks. First, lactic acid is the main byproduct of HA fermentation, and its accumulation causes potent inhibition of cell growth and HA synthesis. Second, fastidious microorganisms require high-cost media from animal origins for growth and fermentation [18,19,20]. Furthermore, HA fermentation by the streptococcal source is frequently accompanied by endotoxin contamination, a restriction that limits the application of the HA product in the biomedical field. Accordingly, recombinant HA production from other microbial platforms has been investigated as an alternative to avoid the safety concerns mentioned above [5,7].

Several reports focused on recombinant HA production with *E. coli* [7,21,22,23,24,25] and *Agrobacterium* sp. [7,26], with HA titers amounting to 3.8 and 0.3 g/L, respectively. *E. coli*-Rosetta host strains (Novagen) allowed researchers to improve low-protein-expression titers caused by codon-usage bias [27].

In addition, the Gram-positive HA (to 6.8 g/L) produced is generally recognized as safe for the GRAS bacterium, *Bacillus subtilis* [28,29], and also for *Corynebacterium glutamicum* [30,31]. Recently, *Bacillus megaterium* was also established as an industrially recognized host. It is non-pathogenic, does not possess any alkaline proteases degrading recombinant gene products, can replicate stably, and can maintain recombinant plasmids even without antibiotic selection [32,33,34]. *B. megaterium* efficiently secretes proteins directly into the culture medium; in some cases, the recombinant protein makes up 30% of the total soluble protein [32,34]. Consequently, *B. megaterium* was used to produce several alpha and beta amylases, which are used for starch modification in the baking industry [33,34], along with penicillin acylase [34,35,36,37] and in aerobic and anaerobic production of vitamin B_12_ [38,39].

Ten enzymes are involved in the HA biosynthetic pathway from glucose (Figure 1). Most of these enzymes possess overlapping functions at the interface between catabolic pathways (glycolysis and pentose phosphate pathway), anabolic production of structural polysaccharides (peptidoglycan and teichoic acid), and the extracellular matrix (HA). Accordingly, it is no surprise to find homologs of most of these enzymes in bacterial species, which do not naturally produce HA. The unique enzyme to HA biosynthetic pathway is the plasma membrane-bound enzyme hyaluronan synthase (*HasA*), encoded by the *hasA* gene. HasA is an integral membrane protein responsible for catalyzing the polymerization of UDP-glucuronic acid and N-acetyl glucosamine and the simultaneous translocation of the growing HA chain across the membrane [40,41].

In this study, we investigated the applicability of four *E. coli* strains (pLys*Y/Iq*, Rosetta2, Rosetta2 (DE3) pLysS, Rosetta-gami B (DE3)pLysS) and *B. megaterium* (MS941) as potential platforms for the heterologous production of HA. Episomal expression systems were constructed for *hasA* along with other genes in the operon involved in the HA biosynthetic pathway. For all host strains, HA production was confirmed by Fourier transform infrared spectroscopy (FTIR), its titer was monitored in different culture media by turbidimetric assay, and its molecular weight was characterized by gel permeation chromatography. Selected recombinant strains were also examined by scanning electron microscopy (SEM).

## 2. Materials and Methods

Following the materials and methods shown in (Appendix A), all chemicals were purchased from Sigma-Aldrich (Labor Chemie GmbH, Steinheim, Germany) or VWR International GmbH (Darmstadt, Germany). Genomic DNA digestion, ligation, and transformation procedures were isolated using standard techniques [42]. Plasmid and PCR purification kits were purchased from Qiagen (Hilden, Germany). Restriction enzymes and T4 DNA-ligase were purchased from New England Biolabs (Frankfurt am Main, Germany). Oligonucleotides were synthesized by Eurofins MWG (Ebersberg, Germany). Phusion Hot Start DNA polymerase was purchased from Thermo Scientific (Dreieich, Germany).

### 2.1. Bacterial Strains, Plasmids Media, and Culture Conditions

*E*. *coli* strains (Appendix A) were cultivated in LB medium [43], while plasmid-harboring *E.coli* strains were grown in triplicates in 50 mL LB medium containing 1% glucose, super optimal broth with catabolite repression (SOC) [44] or TB media [45], supplemented with 100 mg ampicillin/L and 34 mg chloramphenicol/L. Bacterial cultures were grown at 37 °C until an optical density at 600 nm (OD_600_) = 1 was reached, and gene expression was induced by adding β-D-1-thiogalactopyranoside (IPTG) at a final concentration of 0.5 mM in addition to 10 mM MgCl_2_, 2.5 g K_2_HpO_4_/L and 1 g sorbitol/L. Cultures were incubated at 37 °C or 30 °C for 48 h after induction under shaking at 170 rpm in an orbital shaker.

*B. megaterium* MS941 and *B. megaterium* MS941 pre-transformed with T7-RNAP (Mobitec, Munich, Germany) were grown aerobically in either complex or semi-defined media. Complex growth media were either LB-medium supplemented with 1%, 3%, or 5% glucose, sucrose, or TB medium [46]. Semi-defined media were either A5 medium, A5 medium + 4 or A5 medium + MOPSO [37]. For antibiotic selection, media were supplemented with 4.5 mg chloramphenicol/L and 10 mg tetracycline/L. Bacterial cultures were induced at an OD_578_ = 1 by adding 0.25–0.5% xylose in addition to 10 mM MgCl_2_, 2.5 g K_2_HpO_4_/L, and 1 g sorbitol/L, and then the cultures were further incubated at 25 °C for 48 h under shaking at 170 rpm. Genomic DNA was extracted from *S. equi* subsp. *zooepidemicus* ATCC 39920 pre-grown in BHI medium (Oxoid).

### 2.2. Plasmid Construction

Genomic DNA was extracted from *Streptococcus equi* subsp. *zoopedemicus* ATCC 39920 (pre-grown in BHI medium from Oxoid) using the Genomic DNA extraction kit (Qiagen). *hasA*, *hasABC,* and the entire operon *hasABCDE* were PCR-amplified using Phusion polymerase (Thermo Scientific). The primers are listed in Appendix A. DNA was digested using restriction endonucleases purchased from New England BioLabs (NEB; Ipswich; USA) or Thermo Fisher Scientific Inc., Rockford, IL, USA.

PCR reactions of a total volume of 20 μL were prepared to amplify the DNA of interest. For the amplification of DNA fragments more prominent than 1000 bp, the Phusion^TM^ polymerase (Finnzymes; Espoo; Finland) was used for colony PCR, *Taq* DNA polymerase (BioTherm^®^; genecraft; Lüdinghausen; Germany) was used. The Phusion polymerase has a proofreading function that avoids mistakes during amplification. The *Taq* DNA polymerase is known to create one mismatch in 1000 base pairs.

The ‘5’ phosphate groups of the linearized vector were removed before the ligation reaction to avoid the re-circularization of a previously digested DNA vector. The dephosphorylation was achieved by adding 1 unit of calf intestinal alkaline phosphatase (CIP; New England BioLabs; Ipswich; USA) per μg of DNA to the sample immediately after restriction. Incubation at 37 °C for 3 h followed. DNA was purified using the PCR purification kit (Qiagen; Hilden; Germany) following the manufacturer’s instructions. In one ligation reaction, 25–200 ng of plasmid DNA was used. Insert DNA was added in excess (insert-to-vector ratio concerning molar concentrations of 2: 1 to 10: 1), and a reaction buffer 1:10 of the total volume and 200 U of T4 DNA ligase were added to a final volume of 20 μL (New England BioLabs; Ipswich; MA; USA).

*hasA* and *hasABC* were cloned into plasmid pJet1.2 (Qiagen) to form pJet*hasA* and pJet*hasABC*, respectively. *hasA* was subcloned into plasmid pMM1522 to generate plasmid pMM1522*hasA*, and *hasABC* was ligated into pP_T7_ to construct plasmid pP_T7_*hasABC.* Finally, the entire operon was cloned into pP_T7_ to generate plasmid pP_T7_*hasABCDE.* Sequences of all plasmid constructs, Appendix A, were verified using Sanger’s chain-termination-sequencing method.

### 2.3. Bacterial Transformation

Competent *E. coli* cells were prepared and transformed using standard techniques [28]. Competent *B. megaterium MS941* protoplasts were purchased from MoBiTec GmbH; Göttingen, Germany and transformed according to the manufacturer’s guidelines.

For both *E. coli* and *B. megaterium*, only colonies with stable mucoid consistency over several generations were used in HA production (Appendix A).

### 2.4. Growth Experiments and Induction of HA Production

All growth experiments were performed using 50 mL culture medium in 250 mL baffled (1 cm throw) Erlenmeyer flasks in triplicates. HA production by *E. coli* cultures was induced by adding 1 mM–0.05 mM IPTG, while for *B. megaterium,* induction was carried out by adding 0.5–0.1% xylose. In addition to the inducer, 10 mM MgCl_2_, 1 g sorbitol/L, and 2.5 g K_2_HPO_4_/L were added to all culture media [23]. HA was quantified in 1 mL samples withdrawn during the exponential growth phase at different time intervals after induction.

### 2.5. HA Extraction and Quantification in Cell-Free Culture Media

We followed the previously described HA extraction protocol [37]. In brief, 1 mL culture samples were taken at different time intervals in 15 mL Falcon tubes, fivefold diluted with sterile distilled water, and incubated for 10 min with 0.1% sodium dodecyl sulfate (SDS) before being centrifuged at 3000× *g* for 20 min. Supernatants were filtered through 0.22 μm membrane filters to ensure that no bacterial cells were left with 1 M NaCl before adding 2.5 volumes of 100% ethanol. The mixtures were then centrifuged again at 10,000× *g* for 10 min. The clear white HA precipitate was separated and air-dried overnight at room temperature. HA was quantified using the turbidimetric assay [47,48].

### 2.6. Quantification and Characterization of Recombinant HA

For the quantification of HA, a turbidimetric assay was used in which HA formed a complex with CTAB (cetyl-trimethyl-ammonium bromide). The results were not significantly different from those obtained using an ELISA assay (see below). The CTAB assay was recommended due to its robustness, high specificity, and simplicity, and it was suggested to be a superior substitute to the colorimetric carbazole assay, which requires further purification before HA quantification [49].

To test the sensitivity of the assay, we compared the obtained HA levels with those determined in random samples assayed for HA using ELISA (Corgenix, Broomfield, CO, USA), which utilizes an HA-binding protein (HABP) [47,48]. In the ELISA assay, properly diluted HA samples or HA reference solutions were incubated in HABP-coated micro-wells, permitting HA present in the samples to react with immobilized binding protein (HABP). After removing unbound serum molecules by washing, HABP conjugated with horseradish peroxide (HRP) solution was added to the micro-wells to form complexes with bound HA. After another washing step, a chromogenic substrate of tetramethylbenzidine and hydrogen peroxide was added to develop a colored reaction, the intensity of which was measured at 450 nm.

### 2.7. HA Purification for Molecular Weight Determination and FTIR Analysis

Forty-eight hours after induction, cells were separated from the media by centrifugation at 10,000× *g* for 15 min. The clarified spent media were extracted in CHCl_3_ to deproteinize the mixture before quaternary amine detergent precipitation (1% cetylpyridinium chloride [CPC]), and the resulting pellets were washed in water, re-dissolved in 1 M NaCl, the solution was clarified by centrifugation, and the polymer was precipitated by the addition of 2.5 volumes of absolute ethanol. The washing and ethanol-precipitation steps were repeated twice, and the final pellets were then re-dissolved in water, treated with DNase and RNase (1 µg/mL final) for 1 h, then extracted again with CHCl_3_ [49]. The aqueous phase was harvested and eluted with 500 µL Milli-Q water system (Millipore) in an Amicon column (MW cut off >10,000 Da) and was centrifuged at 7000 rpm for 20 min to remove traces of CPC and protein. The solution left above was used for molecular weight determination.

### 2.8. Molecular Weight Determination of HA

The molecular weight of HA was determined using a Waters 515/2410 Gel Permeation Chromatograph (GPC, Waters, Milford, MA, USA) with an ultra-hydrogel column calibrated with polyethylene glycol standards and a series 2410 refractive index detector. Mobile phase: water, sodium nitrate (0.10 M). Solvent: water, sodium azide 0.05%; flow rate: 1 mL/ min; temperature: 25 °C.

### 2.9. Scanning Electron Microscopy of HA Capsules

Dried bacterial cultures were processed by suspending them in 3% glutaraldehyde of 0.1 M phosphate buffer for 1 h, pH 7.2. After washing three times, post-fixing was conducted in 1% osmium tetroxide (in H_2_O) for 1 h. Afterward, the cells were dehydrated in 30%, 50%, 75%, 90%, and 2 × 100% for 5 min each. A 0.2-micron polycarbonate filter was used for purifying the dried bacteria, sucked by a vacuum pump in 100% ethanol [50]. Dried bacterial cultures were coated with metal stubs of the thin layer of heavy metal to increase the secondary electron signal (usually gold or gold palladium). The dried bacterial cultures were imaged in a JCM-5700 Scanning Electron Microscope (JEOL USA, Peabody, MA, USA), equipped with a mobile biological containment (Dycor Technologies Ltd., Edmonton, AB, Canada). Images of the dried bacterial cultures were taken after they had been processed under high vacuum at 6 kV, with an 8 mm working distance and a 30 μm objective lens aperture. A secondary electron detector was the tool used to collect the images, with which the acquisition time per image was 160 s, and each image was 2560 × 1920 pixels. Images of the ionic-liquid-stained samples were obtained using the above-noted settings, except that the acceleration voltage was adjusted to 4 kV. SEM images were recorded at magnifications ranging from 3000× to 20,000×.

### 2.10. Characterization of HA Using FTIR

The characterization of representative HA samples [51] was performed using FTIR spectrophotometry (Jasco, Hachioji, Japan), and the spectra were compared to a standard sample of HA (Sigma Aldrich Chemicals, Burlington, MA, USA) between 4000–400 cm^−1^ (2.5–25 μm). All organic molecules can absorb FTIR radiation between 4000 cm^−1^ and 400 cm^−1^, corresponding to energy absorption between 11 kcal/M and 1 kcal/M. This amount of energy initiates transitions between vibrational states of bonds contained within the molecule.

### 2.11. Data Processing and Statistical Analysis

Statistical analysis was performed using the statistical program SPSS V.17 [52]. Data are represented as means ± SEM. To compare differences between groups’ odds ratios, a nonparametric Student’s *t*-test (Mann–Whitney), Wilcoxon signed-rank test, and nonparametric one-way ANOVA (Kruskal–Wallis) were used. A two-tailed *p*-value ≤ 0.05 was considered statistically significant in all statistical tests.

## 3. Results

### 3.1. HA Production with E. coli

The *hasA* gene from *Streptococcus equi* subsp. *zooepidemicus* ATCC 39920 was amplified and ligated into plasmid pMM1522 under the control of a xylose-inducible promoter. *E. coli* Rosetta2 was transformed with the resulting plasmid pMM*hasA,* and the most HA-producing colony of a stable mucoid consistency was chosen. Unexpectedly, HA production by this strain did not exceed 8.8 ± 0.8 mg/L after 48 h of incubation at 37 °C in an LB medium. Accordingly, we explored the strategy of cloning *hasABC,* and the entire *has* operon *hasABCDE* under the control of the firm and tightly regulated T7 promoter. We used *E. coli* INV alpha F’, which is known to harbor a soft copy of the foreign plasmid [53] for *hasABC* and *hasABCDE* cloning; this, together with the addition of glucose (1%) to the growth medium, seems to completely shut off the leaky expression of the *hasABC* and overcome potential gene toxicity. *hasABC* from *S. equi* subsp. *zooepidemicus* was successfully ligated into plasmid pP_T7_, resulting in plasmid pP_T7_*hasABC*. *E. coli LysY/Iq* cells being transformed with pP_T7_*hasABC*. This *E. coli* strain carries the T7 RNA polymerase gene under the control of LysY, a variant of T7 lysozyme lacking amidase activity; thus, cells are less susceptible to lysis during induction, according to the manufacturer [54]. The plate-assay method was used to choose the mucoid colony that grew the most in the presence of IPTG and ampicillin. At 1 mM IPTG, the growth was prolonged, and the HA titer was very low. By reducing the IPTG concentration to 0.5 mM and supplementing the LB medium with 10 mM MgCl_2_, 2.5 g K_2_HPO_4_/L, and 1 g sorbitol/L, the HA titer was 78.3 ± 4.4 mg/L after 48 h at 37 °C, which was still lower than the HA concentrations that were reported earlier [19,33]. After induction, the titer of the polymer increased to 208.3 ± 10.9 mg/L after 48 h with a temperature of 30 °C (Figure 2). Therefore, since the *hasA*BC gene was introduced into *E. coli lysY/I^q^* cells was not codon-optimized, we attributed the low HA productivity by this host strain to a bias in codon usage. Accordingly, we tested the use of the *E. coli* strains Rosetta2 (DE3) plysS and Rosetta-gami B (DE3)pLysS, which both enhanced the expression of heterologous (non-codon-optimized) genes. Both strains were grown in LB media supplemented with 1% glucose, and the HA titer was determined at different time points (Figure 3A). Under these conditions, Rosetta-gami B(DE3)pLysS exhibited a slightly higher HA titer, probably due to enhanced disulfide bond formation and improved protein folding [54,55]. Further improvements were attained when SOC and TB media were used after 48 h of incubation, resulting in HA concentrations of 346.7 ± 3.3 mg/L and 500 ± 11.4 mg/L, respectively (*p* < 0.001; Figure 3B). No detectable effect was observed in our study when lysozyme was applied at a concentration of 500 U/mL, as recommended for *Streptococcus* [37]. The lysozyme encoding plasmid pLysS in Rosetta-gamiB cells as a control system to prevent leaky expression reported for T7 promoter systems has led to further suppression of the HA expression. In this respect, adding lysozyme to the TB medium has led to declining HA production compared to the TB medium.

The entire operon *hasABCDE* from *S. equi* subsp. *zooepidemicus* was successfully ligated into plasmid pP_T7_ to construct plasmid pP_T7_*hasABCDE.* After verification of the sequence by Sanger sequencing, E*. coli* Rosetta-gamiB (DE3) pLysS cells were transformed, and transformants were grown in TB medium. The HA titers reached 585 ± 2.9 mg/L, significantly higher than the HA titer obtained for pP_T7_*hasABC* (Figure 4).

*E. coli* JM109 was used for the co-expression of *hasA* from *Pasteurella multocida* and the UDP-glucose dehydrogenase gene *(kfiD)*, a *hasB* analog from *E. coli* K5. In shake flasks, the HA titer of this strain was at 500 mg/L [19,22]. In a latter study, glucosamine was added to the feeding solution to de-couple growth and capsular polysaccharide (CPS.) synthesis, and it was determined that insufficient precursor supply was detrimental to HA synthesis [20,21].

### 3.2. HA Production with B. megaterium

A potential *B. megaterium* expression platform was suggested for recombinant HA production due to its ability to secrete recombinant proteins directly into the culture medium. Plasmid pMM1522-*hasA* was transformed into *B. megaterium* MS941 protoplasts pre-transformed with pT7-RNAP. Cells were grown to an OD_578_ = 1 before being induced using a mixture containing 0.25% xylose, 10 mM MgCl_2_, 10 mM K_2_HPO_4,_ and 1 g sorbitol/L (see Methods). The presence of sorbitol has been shown to promote the mucoid phenotype of the transformed colonies. The productivity of the HA was only 50 ± 5.7 mg/L after 48 h. The low HA titer was expected when only *has*A was expressed, implying a limitation of essential HA precursors, as reported for *B. subtilis* cells expressing *hasA* only [22]. Subsequently, plasmid pP_T7_*hasABC* was used, and the LB culture medium was supplemented with 1%, 3%, or 5% glucose (Appendix A).

For the three used glucose concentrations (Figure 5A): After induction with 0.5% xylose, the HA titers reached 1023.3 ± 39.2 mg/L, 1200 ± 57.7 mg/L, and 1833 ± 60.1 mg/L, respectively. A further increase in HA productivity was achieved by the LB medium supplemented with different sucrose concentrations (1%, 3%, and 5%) to reach 1403 ± 8.8 mg/L, 1823.3 ± 14.5 mg/L, and 2116.7 ± 44 mg/L, respectively, while in the TB medium, the HA titer was 2016 ± 44 mg/L (Figure 5B).

We also explored using the semi-minimal growth media A5, A5 + 4, and A5 supplemented with the buffering agent MOPSO (2-hydroxy-3-morpholinopropanesulfonic acid; 2-hydroxy-3-morpholinopropane-1-sulfonic acid) as a nitrogenous base for optimal pH constant for bacterial growth. The incubation of *B. megaterium* MS941 protoplasts pre-transformed with pT7-RNAP in these media resulted in HA titers of 1900 ± 25.1 mg/L, 1873.3 ± 14.5 mg/L, and 1988.3 ± 19.6 mg/L, respectively (Appendix A). A5 + MOPSO medium supported the highest HA productivity compared to the other semi-minimal media (Figure 5B).

As *B. megaterium* MS941 (pP_T7_*hasABC*) cultured in LB medium supplemented with 5% sucrose or in A5 medium + MOPSO exhibited by far the highest HA titers in complex media and semi-minimal media. These media were also used for growing *B. megaterium* MS941 harboring the entire HA operon (pP_T7_*hasABCDE*). The HA titer was 2350 ± 28.9 mg/L in LB medium containing 5% sucrose and 2476.7 ± 14.5 mg/L in A5 medium + MOPSO (Figure 5B). The higher titers indicated that the co-expression of *hasDE* relieved the limitation in HA precursors when only a part of the operon was used.

Finally, the HA production by *B. megaterium* was almost fourfold higher than the highest HA levels produced by any of the *E. coli* strains investigated (Figure 6).

### 3.3. Molecular Weight and Polydispersity Index (PDI) of Recombinant HA

Table 1 summarizes the average MW and PDI values for the HA products, which were obtained in all experiments. For *E. coli* strains, the highest HA MW values were obtained for cells expressing the entire operon (pP_T7_*hasABCDE*) and grown in TB medium (8.3 × 10^5^ ± 78.4 Da), and the lowest MWs for cells expressing *hasA* only (pMM1522*hasA*) and grown in LB medium supplemented with 1% glucose (1.2 × 10^5^ ± 83.5 Da). In contrast to previous reports, the presence of *hasE (pgi)* seems to have little or no significant effect on HA MW [19]. However, a good comparison may not be possible due to the different methodologies used in MW determination. The PDI levels followed a reverse pattern, lowest for the entire operon (1.3) and highest for *hasA* (1.5). The addition of lysozyme resulted in lower molecular weights and higher PDI.

For *B. megaterium,* HA MW obtained by overexpression of *hasA* (pMM1522*hasA*) was 6.8 × 10^5^ ± 28.4 Da, almost sixfold higher than the corresponding *E. coli* levels expressing the same gene, while the overexpression of *hasABC* (pP_T7_*hasABC*) resulted in HA MW ranging from 9.9 × 10^5^ ± 6.5 to 1.2 × 10^6^ ± 30.5 Da. In semi-minimal media, (A5, A5 + 4, A5 + MOPSO) HA MW was 1.2 × 10^6^, 1.2 × 10^6,^ and 1.4 × 10^6^ Da, respectively, and PDI was 1.2. Thus, A5 + MOPSO showed the highest MW observed with a minor PDI. It was also encouraging to overexpress *hasABCDE* in the same medium where the observed HA MW was 1.9 ± 52 × 10^6^ Da and PDI was 1.2. The presence of *hasDE* genes resulted in significant differences in HA MWs. It increased the flux of the gene product, UDP-GlcNAc, which is a crucial player in determining HA MW [39].

However, the HA MWs obtained in the present study were lower than those previously reported for *B. subtilis* expressing *hasA* from *P. multocidia* in combination with the *B. subtilis* analogs of *hasB* (*tua D*), *hasC* (*gtaB*) and *hasE* (*pgi*:glucose-6-phosphate isomerase) [23]. In summary, *B. megaterium* expressing recombinant *has* genes produced a higher quality of the polymer (higher MW and lower PDI) than *E. coli* strains to harbor identical plasmids. Whereas the availability of UDP-GlcNAc, the product of *has*DE genes, resulted in higher HA MW in *B. megaterium*, *has*DE expression in *E. coli* did not lead to the same effect. Furthermore, a scanning electron microscope was used for HA capsule detection to investigate the potentiality of *B. megaterium* over *E. coli* in HA production.

### 3.4. Scanning Electron Microscopic (SEM) Examination

SEM analysis showed that induced *E. coli* Rosetta-gami B(DE3)pLysS cells carrying pP_T7_*hasABCDE* formed a capsule (Figure 7A). It disappeared after incubation with hyaluronidase (0.1% in PBS for 30 min at room temperature). It indicated its apparent nature as HA (Figure 7B). *B. megaterium* harboring pP_T7_*hasABCDE* and T7RNAP, on the other hand, did not accumulate capsules and seemed to shed HA into the medium, after being incubated with hyaluronidase under the same conditions (Figure 8A,B). *Corynebacterium glutamicum* has been also reported to shed the HA polymer into the culture medium without forming a distinct capsule [23].

Capsule disruption is an essential step for HA quantification that should precede the assay procedure. The low HA titers obtained for *E. coli* compared to *B. megaterium* likely result from inefficient capsule dissolution.

### 3.5. FTIR Spectrum of HA Produced by Recombinant E. coli and B. megaterium Strains

HA samples prepared as described in the “Methods” section showed an FTIR spectrum similar to the HA standard (Appendix A). The positions of the six peaks, in terms of wave number (cm^−1^), of both the reference standard and HA produced by pP_T7_*hasABCDE E. coli* Rosetta-gami B pLysS and pP_T7_*hasABCDE B. megaterium*, under optimal conditions, were identical (Figure 9A,B). It indicated the efficiency of the purification method employed in this study.

## 4. Discussion

Our attempts to obtain stable clones carrying *hasAB* or *hasABC* in pMM1522 were unsuccessful, probably because of the leaky nature of the xylose promoter in *E. coli*. It is based on *xylAB* genes, as reported earlier [32]. All subsequent culturing conditions for *E. coli* were performed at a temperature of 37 °C after induction. It was previously indicated that temperature switches improve the HA titer in *Streptococcus equi* subsp. *zooepidemicus*, probably because the slower growth rate provides enough time for correct protein folding via the weakening of the glycolytic process, enhancing the HA synthesis process, which reduces the biomass-formation rate [55]. HA productivity titers in *E. coli* strains Rosetta2 (DE3) plysS and Rosetta-gami B (DE3)pLysS were higher than those previously reported for the same media [18,34]. The improved productivity is probably attributed to the solid T7 promoter and the buffering of the TB medium, which could maintain the neutral pH crucial to HA production [56]. However, lysozyme has been previously suggested to increase HA productivity of *Lactococcus lactis* when applied 24 h after induction [19]. This suppressive effect of lysozyme on HA production has similarly been reported for *Streptococcus equi* [40], indicating the expression of the entire operon *hasABCDE* indeed increased the availability of HA precursors UDP-GlcA and UDP-N-GlcNA, in agreement with previous recommendations that emphasize the importance of the expression of *hasE (pgi)* together with *hasABC* for optimized HA productivity [19,39]. HA production in *E. coli* was significantly affected by factors such as culture temperature, the codon usage of the heterologous genes, promoter strength, and the specific type of *E. coli* host strain. These factors can influence the solubility of the expressed proteins and control the HA biosynthesis enzymes (Figure 1). However, the obtained HA titers in *E. coli* were still significantly lower than those reported for the Gram-positive bacterial hosts [7,23,40,41].

Although TB medium resulted in a relatively high HA titer in *B. megaterium*, HA production was not observed until 16 h from the onset of growth. The enhancing effects of glucose and sucrose on HA productivity have been previously reported in several studies [16,54,55,56]. The solid buffering capacity of MOPSO seems to play a significant role in optimizing the culture conditions by maintaining a neutral pH throughout the entire growth period, eventually leading to improved HA production [44,57,58]. It is noteworthy that the advantages of *B. megaterium* over *Lactococcus lactis* in producing HA are direct to the choice of the growth media. *Lactococcus lactis* typically produces HA in a capsulated form, which requires more time and cost for HA purification. *B. megaterium* can be a better industrial host strain for HA production. The vector systems used in our study were commercially available. We also concluded from our study that HA production needs well-controlled expression systems, as *hasA* is toxic, and it is challenging to clone genes when the expression system is not well-controlled. We have also tried to use the shuttle vectors *E. coli*/*Corynebacterium glutamicum*, but the cloning failed due to the leaky expression of the pTac system used. In conclusion, it would be better to use the pPt7 system when we want to produce HA by *Corynebacterium glutamicum*; this is our outlook for the future.

From our study, we suggest that the following adjustments have contributed to the high HA productivity obtained with *B. megaterium*, whether from *hasABC* or *hasABCDE* expression; the different optimization factors between *B. megaterium* and *E. coli* showed a potential influence on HA-production characteristics:The initial lack of cell walls in the *B. megaterium* protoplasts MS941 favors HA production by limiting the competition between HA production and cell-wall biosynthesis.Decreasing the temperature after induction from 30 °C to 25 °C helps decrease the growth rate, thus directing the carbon flow from the available substrates (glucose or sucrose) to HA production.Using the potent buffering agent MOPSO in semi-minimal media (A5 + MOPSO) supplemented with 5% sucrose was essential.

The high productivity of HA by *B. megaterium* compared with *E. coli* may be attributed to the ability of *B. megaterium* to maintain several stable extra-chromosomal DNA elements replicating in parallel [59,60]. It has been crucial to express more genes of the HA biosynthetic pathway without encountering plasmid-instability events. In addition, UDP-GlcNAc is limited in *E. coli,* and this substrate plays an essential role in the HA titer.

The molecular weight (MW) of HA is an essential criterion in the characterization of the polymer that defines its physiochemical and biological properties and applications. Although the HA biosynthetic mechanism is well established, little is known about the molecular mechanisms that control chain termination. Hence, the MW of produced HA is true not only for hyaluronan synthases but also for other β-polysaccharide synthases, e.g., cellulose, chitin and 1,3-betaglucan synthases [44]. High MW HA polymers, capable of reaching an HMW of 10^8^ Da, are suitable as dermal fillers, anti-angiogenic agents, and potentiality as immunosuppressives via fibrinogen binding, and enhance inflammatory cytokines as well as stem-cell mitigation. Medium-size hyaluronan chains (between 2 × 10^4^ and 10^5^ Da) are involved in ovulation, embryogenesis, and wound repair. Oligosaccharides with 15 to 50 repeating disaccharide units (between 6 × 10^3^ and 2 × 10^4^ Da) are anti-inflammatory, immuno-stimulatory, and angiogenic, while small hyaluronan oligomers (from 400 to 4000 Da) are anti-apoptotic and inducers of heat-shock proteins [2,61,62].

Another critical parameter is the polydispersity index (PDI) of the polymer. The reported molecular weight of isolated HA polymers may be under-evaluated depending on the extraction, isolation, and analysis methods used. Consequently, the molecular weight of HA is experimentally determined as displaying the polydispersity(*Mw/Mn*) of the polymer. It depends on both the method of extraction and the method of analysis used [63].

In the present study, HA MW was determined using gel permeation chromatography (GPC), which has been proven to be the most accurate method for MW determination. In this context, it is worth saying that PDI is a pivotal factor in determining the appropriate applications of HA, with a low PDI suggesting a priority in medical/industrial applications due to its few discrepancies or narrow size distribution, that is, a low heterogeneity index for the HA [48].

In conclusion, there are a lot of research opportunities and challenges in recombinant HA. HA production can be achieved by Gram-negative (*E. coli*) or Gram-positive (*B. megaterium*) bacterial platforms. However, the superiority of Gram-positive bacteria (*B. megaterium*) can be attributed to their capacity for HA-production delivery to the medium without displaying a capsulated form. However, the media composition, the buffering system (pH neutralization), and temperature could play a critical role in the chosen plasmid system, whether Ptac or pP_T7_. The study’s limitations were the choice of stable mucoid colonies and the possibility of using *B. megaterium* protoplasts at an industrial scale, as well as media optimization and the use of low-cost media and those suitable for increasing the HA yield. More efforts are recommended to reach optimal and stable HA production, starting from genomics (gene-code optimization) to proteomics (controlling HA synthase genes, together with other genes involved in HA biosynthesis), and from there to the proper choice of bacterial strains.

## Figures and Tables

**Figure 1 microorganisms-10-02347-f001:**
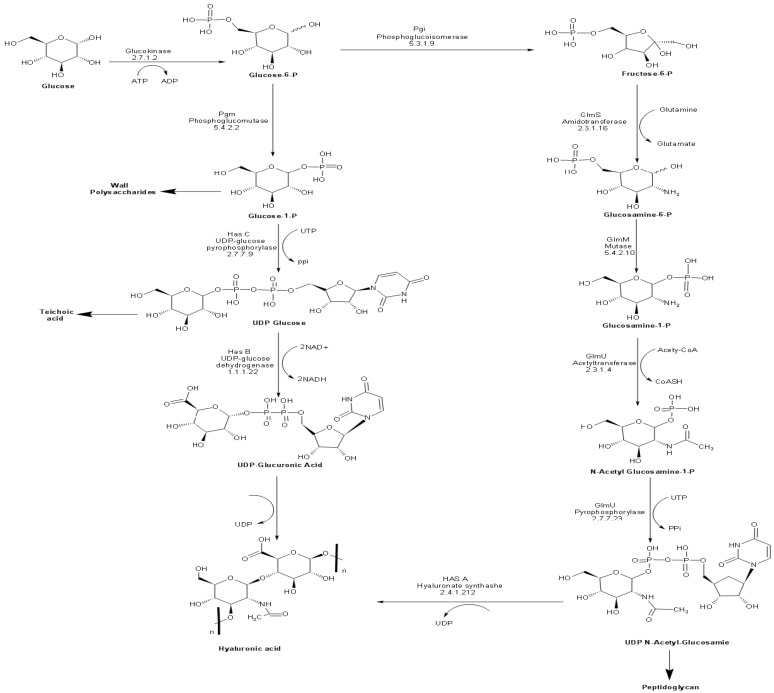
Biosynthetic pathway of HA in *S. equi* subsp. *zooepidemicus*.

**Figure 2 microorganisms-10-02347-f002:**
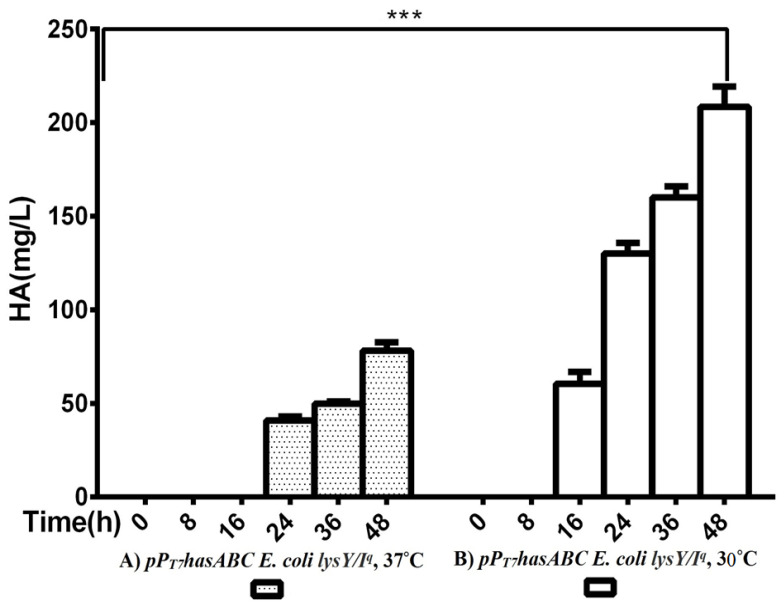
Comparison of HA productivity between *pP_T7_hasABC *E. coli* lysY/I^q^* at (**A**) 37 °C and at (**B**) 30 °C, (***): *p* ˂ 0.0001.

**Figure 3 microorganisms-10-02347-f003:**
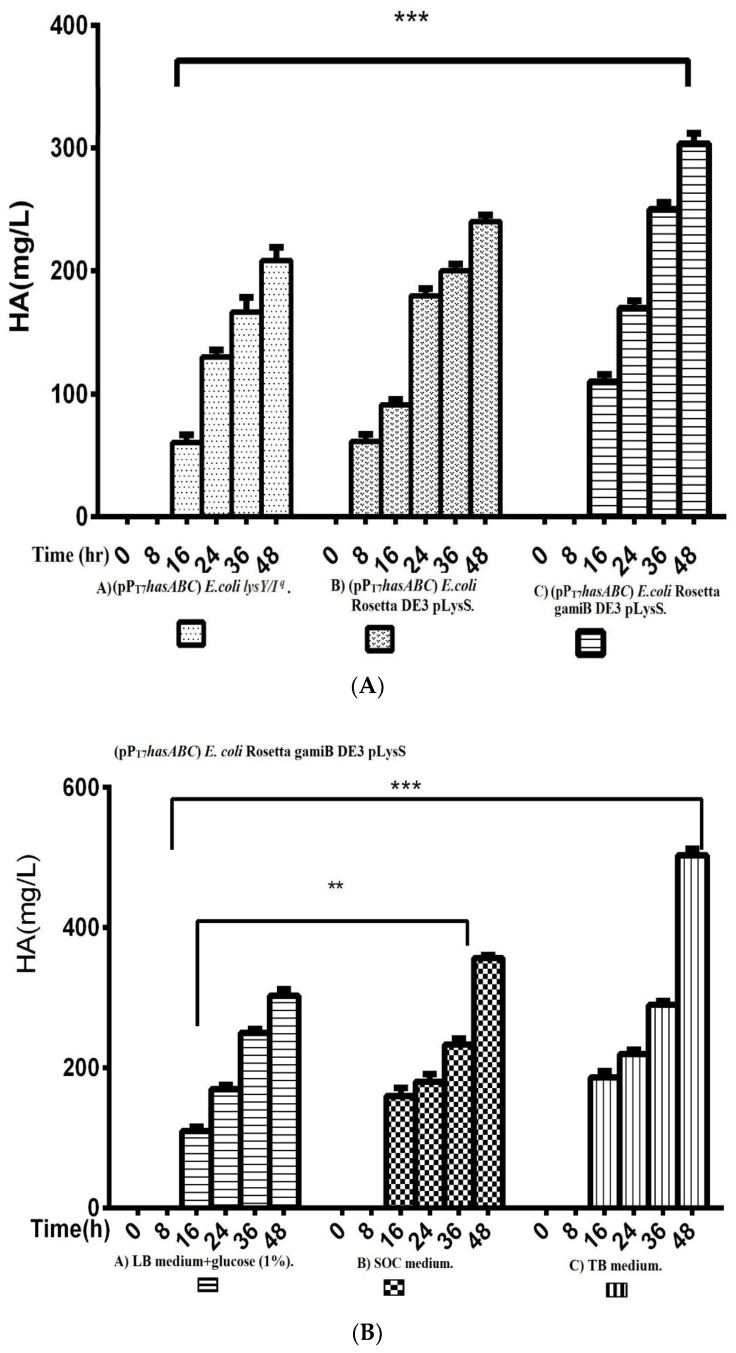
Comparison of HA productivity presented by (Figure 3**A**); A) (pP_T7_*hasABC*) *E.colilysY/I^q^*, B) Rosetta DE3 pLysS, and C) Rosetta gamiB DE3 pLysS, (***): *p* ˂ 0.0001. (Figure 3**B**) Comparison of HA productivity by *E. coli* Rosetta gamiB DE3 pLysS transformed with (pP_T7_*hasABC*) in A) LB medium + glucose (1%), B) SOC medium, and C) TB medium, (**): *p* < 0.001, (***): *p* ˂ 0.0001.

**Figure 4 microorganisms-10-02347-f004:**
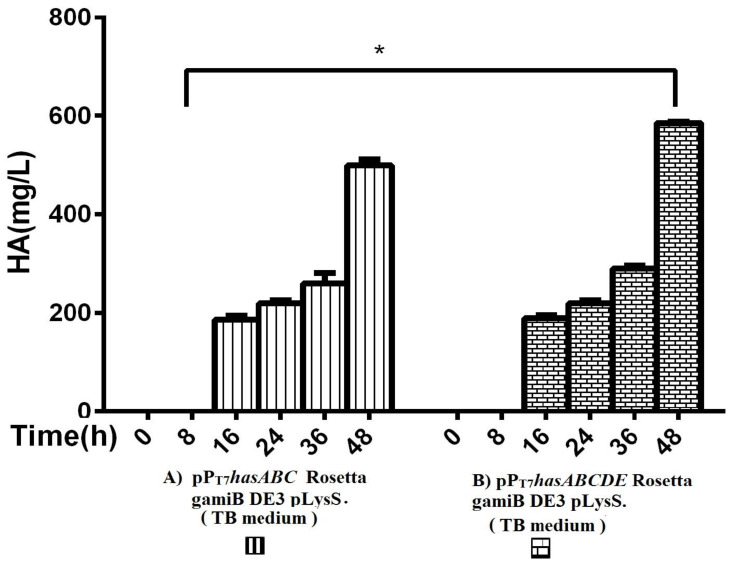
Comparison of HA productivity in TB medium between (**A**) pP_T7_*hasABC* Rosetta gamiB (DE3) pLysS and (**B**) (pPT_7_*hasABCDE*) Rosetta gamiB (DE3) pLysS, (*): *p* ˂ 0.05.

**Figure 5 microorganisms-10-02347-f005:**
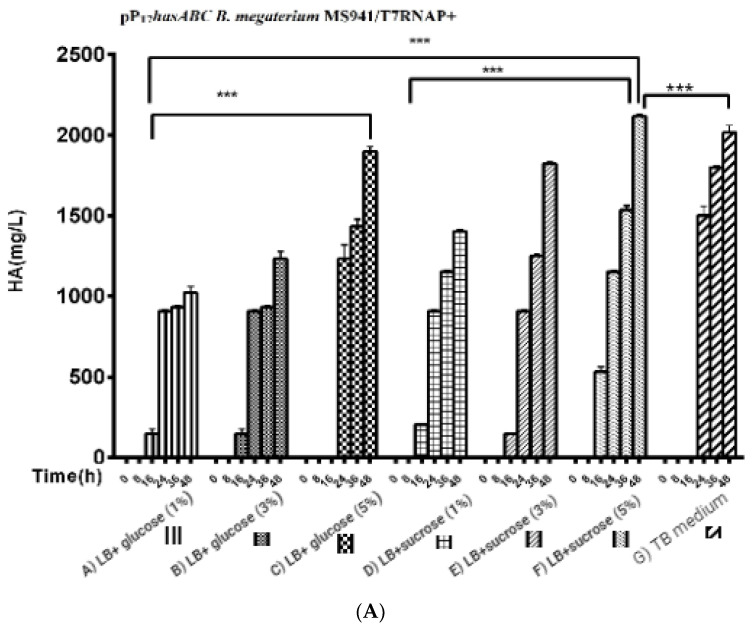
Comparison of HA production presented by (Figure 5**A**); pP_T7_*hasABC B. megaterium* MS941 pre-transformed with T7RNAP in complex media of A) LB + glucose (1%), B) glucose (3%), C) glucose (5%), D) LB + sucrose (1%), E) LB + sucrose (3%), F) LB + sucrose (5%), and G) TB medium, (***): *p* < 0.0001. (Figure 5**B**): Comparison of HA production by Pp_T7_*hasABC B. megaterium* MS941 pre-transformed with T7RNAP in semi minimal media A) A5, B) A5 + K, and C) A5 + MOPSO, (*): *p* ˂ 0.05.

**Figure 6 microorganisms-10-02347-f006:**
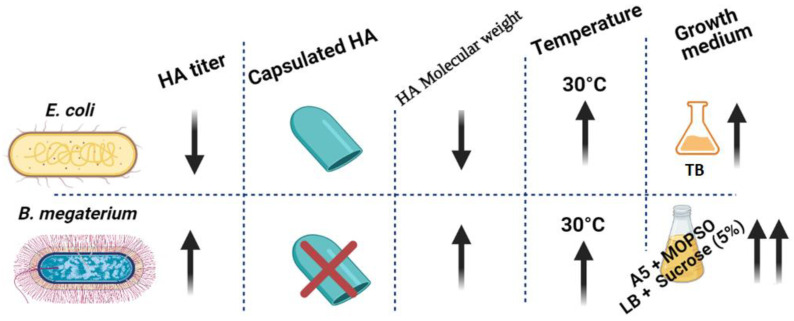
Graphical conclusion of the superiority of HA productivity in *B. megaterium* over *E. coli*. HA productivity using *B. megaterium* platform showed higher HA titer with lower polydispersity and no capsulated HA in the culturing media of A5 + MOPSO and LB + sucrose (5%) (up-arrow represents “increase”, down-arrow represents “decrease”).

**Figure 7 microorganisms-10-02347-f007:**
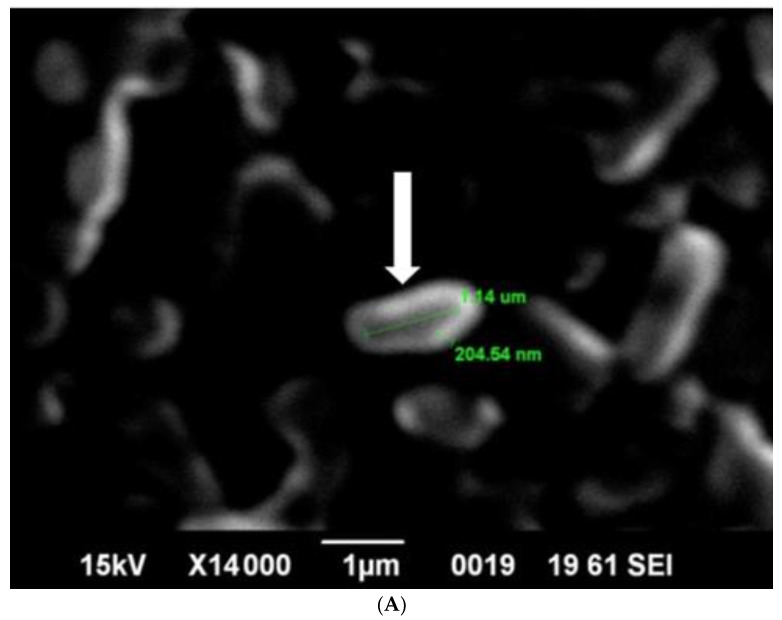
(**A**): SEM photo of induced pP_T7_*hasABCDE E. coli* Rosetta-gamiBpLysS. HA capsule was found intact in the cell, with white arrow. (**B**): SEM: Photo of induced pP_T7_*hasABCDE E. coli* Rosetta-gamiBpLysS after incubation for 20 min with hyaluronidase enzyme dissolved in phosphate buffered saline.

**Figure 8 microorganisms-10-02347-f008:**
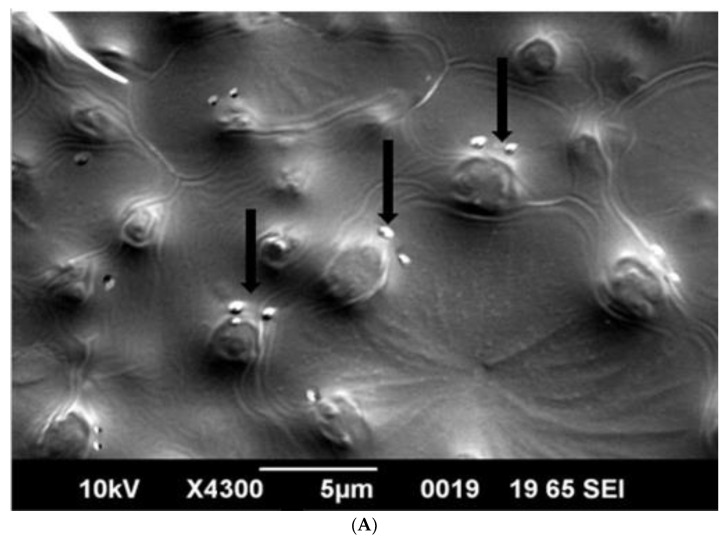
(**A**): SEM photo of induced pP_T7_*hasABCDE B. megaterium* MS941. It was clear that HA particles were outside the protoplasts and do not form a distinct capsule. They appeared as white particles (black arrow). (**B**): SEM photo of induced pP_T7_*hasABCDE B. megaterium* MS941 after incubation for 20 min with hyaluronidase enzyme dissolved in phosphate-buffered saline.

**Figure 9 microorganisms-10-02347-f009:**
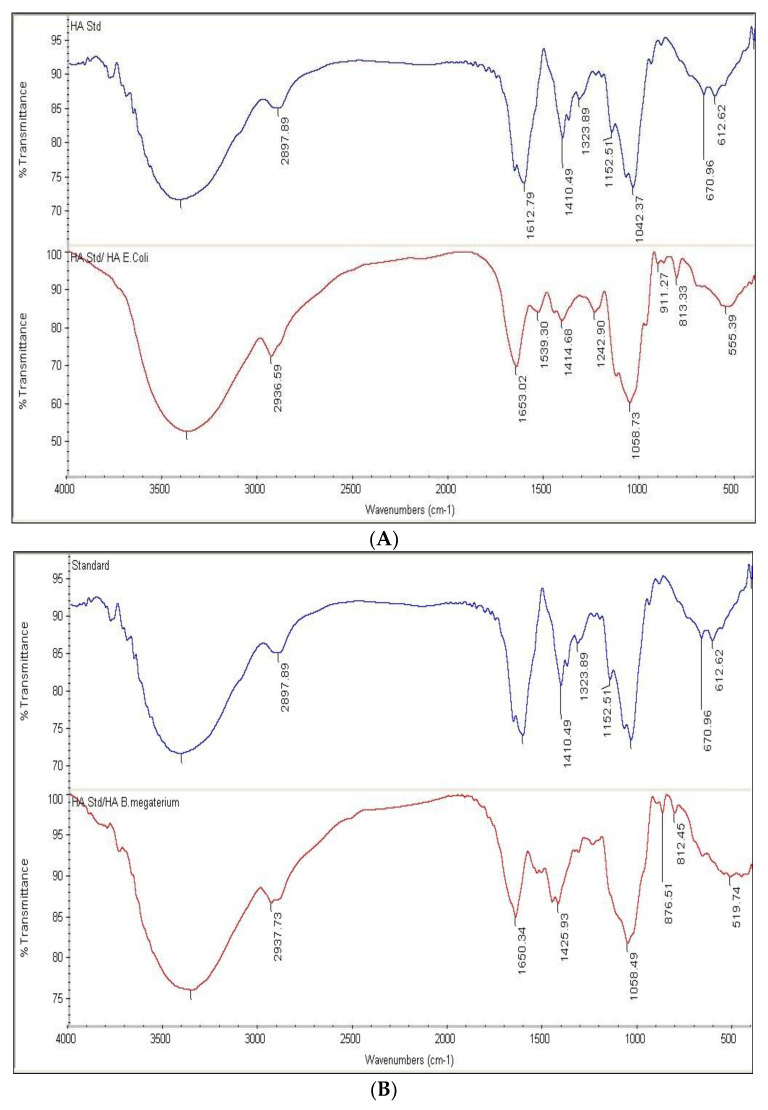
(**A**): FTIR of HA standard versus HA extracted from pP_T7_*hasABCDE E. coli* Rosetta-gamiBpLysS. (**B**): FTIR of HA standard versus HA extracted from pP_T7_*hasABCDE B. megaterium*.

**Table 1 microorganisms-10-02347-t001:** Summary of HA quantification and molecular weight determination in all strains used in this study.

Construct	Strain	Medium	Mwt (Da)	PDI
pMM1522*hasA*	*E. coli* Rosetta 2	LB + glucose 1%	1.2 × 10^5^ ± 83.5	1.5
	*B. megaterium*	LB + glucose 1%	6.8 × 10^5^ ± 28.4	1.5
pP_T7_*hasABC*	*E. coli LysY/Iq*	LB + glucose 1% at 37° C	n.d	n.d
	*E. coli* Rosetta B gami DE3pLysS	L LB + glucose 1%	1.9 × 10^5^ ± 14.5	1.5
		SOC	5.4 × 10^5^ ± 14.5	1.4
		TB	8 × 10^5^ ± 35.8	1.3
		LB + lysozyme	5 × 10^5^ ± 33.3	1.7
	*B. megaterium* MS941, transformed with T7NAP	LB + glucose 1%	9.9 × 10^5^ ± 6.5	1.5
		LB + glucose 3%	9.6 × 10^5^ ± 18.9	1.5
		LB + glucose 5%	1.2 × 10^6^ ± 30.5	1.5
		LB + sucrose 1%	9. 2 × 10^5^ ± 30.5	1.6
		LB + sucrose 3%	9.8 × 10^5^ ± 32.5	1.6
		LB + sucrose 5%	1.2 × 10^6^ ± 38	1.6
		TB	9.9 × 10^5^ ± 56	1.3
		A5 + 4	1.1 × 10^6^ ± 47	1.2
		A5 + K	1.2 × 10^6^ ± 25	1.2
		A5 + MOPSO	1.4 × 10^6^ ± 30	1.2
pP_T7_*hasABCDE*	*E. coli* Rosetta B gami DE3pLysS	TB	8.3 × 10^5^ ± 78.4	1.3
	*B. megaterium* MS941transformed with *T7RNAP*	LB + sucrose 5%	1.7 × 10^6^ ± 18	1.6
		A5 + MOPSO	1.9 × 10^6^ ± 52	1.2

## Data Availability

All Data are contained within the article and Appendix A.

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
