# Peer review of "The Superiority of Bacillus megaterium over Escherichia coli as a Recombinant Bacterial Host for Hyaluronic Acid Production"

_microorganisms, 2022, doi:10.3390/microorganisms10122347_

Round 1

Reviewer 1 Report

In the present study, different bacterial host strains including gram-negative and gram-negative strains were tested for the production of recombinant hyaluronic acid. Furthermore, the fermentation conditions were optimized for high HA production. And the molecular mass of representative HA samples was characterized as well. In general, the overall project is well-designed. I only have some small suggestions for the authors to revise and explain.

1. The authors used five types of E. Coli for the HA production test. However, only one type of Gram-positive Bacillus megaterium was used for the HA production test. Why do the authors design them in this way? Because some gram-positive bacteria, including Corynebacterium glutamicum, can be engineered to produce HA.

2. Under different media conditions, strains with the same genotype produce various molecular weights of the HA products as shown in Table 1. What are the possible reasons? I suggest that the authors add them to the discussion section.

Minor suggestions:

1. Some words missing in the Purpose column in Table (S4): List of oligonucleotides.

2. The name of the strain should be italicized in the Abstract section.

3. Figure 9A and Figure 9B should be combined into one figure using the same processing method in Figure 5, one on the left and one on the right.

Author Response

Dear reviewer,

Kindly find our reply, attached below.

Best regards

The authors

Reviewer 2 Report

The manuscript by Nasser et al describes the biotechnological production of hyaluronic acid in E. coli and B. megaterium bacteria hosts. Authors test different plasmid constructs, harbouring the hasA gene (HA synthase) or the HA operon ABCDE, in both hosts and determine the yield and features of the HA recovered. They also screen for media composition to maximise the production of HA. Under the most productive host and conditions, authors obtained a yield of over 2g/L of HA, with a MW of 105 to 106 Da and a polydispersity index <2. Heterologous production of HA in bacteria is not novel and their reports of higher HA titers higher than the ones reported in this manuscript. Nevertheless the current study is interesting and presents two novelties over previous studies: on one hand, the expression of the HA operon in the host bacteria and, on the other hand, the use of the gram positive Bacillus megaterium as a production host of HA. Therefore, in my opinion the current manuscript deserves publication in Microorganisms.

there are a few points which I would like to suggest authors to think about in order to improve the manuscript: 

-authors chose to structure the manuscript with a combined Result and Discussion section, which I think it is fine. However, I miss a final conclusion section, which could be the last paragraph of the Results and Discussion section, where they can summarise all the observations and data presented in the results section, and put them into context with current production methods of HA (how does the yield compare with streptoccal fermentation and heterologous production systems?). In addition, they can provide critical analysis about the limitations of the expression system that they tested and how amenable it would be for large scale fermentation. they can also suggest interventions that could improve the productivity of their expression system.  Authors can refer to Fig. 6 in this paragraph. Regarding Figure 6, I am not sure if the gaps under Temperature and Growth medium (missing arrows for B. megaterium and E. coli, respectively) are intended or they are errors. In addition, the arrows under polydispersity contradict the text of the figure legend. In addition, the last sentence of Figure 6 legend seems not finished. 

-I think that a potential reader would appreciate some help in interpreting the SEM pictures presented in this manuscript. For example, authors can use arrowheads to point to HA capsules, draw the contour of cells, etc. 

Typos and other comments

-"determined", on the 7th line of second paragraph of page 4

-remove unnecessary bracket in line 3, third paragraph, page 5

-page 6, second paragraph, I think the text refers to Figure 5B instead of 3B

-page 6, fourth paragraph, spell out the genus of C. glutamicum

-page 11, last paragraph, I do not think the word "neutralized" applies to the addition of 1M NaCl to a solution. 

 -Figure S1: there is a discrepancy between the media described in the figure legend and the media shown in the graph. 

-Table S1. Please revise spelling of chloramphenicol

-Table S3. Please revise spelling of chloramphenicol and ampicillin. 

Author Response

(The authors gave the same response as above.)

Reviewer 3 Report

This paper describes production of recombinant hyaluronic acid by a few E coli strains and one gram positive strain Bacillus megaterium. Specifically, the paper shows how B megaterium could yield higher and more purer titers of HA. The manuscript provided to me lacks line numbers and hence I am unable to provide detailed line by line corrections for spelling or grammar. My comments are listed below. 

1.  The authors must discuss the rationale behind first transforming only hasA, AB or ABC in some of their experiments before moving to the entire operon. As the manuscript currently stands, the significance of not straightaway working with hasABCDE is unclear. 

2. Results and Discussion section needs to be separated. Large chunks of information in this section is related to discussion and takes the focus away from the experiments / data in question. For eg: The paragraph about E coli JM109 does not belong to the result section. Similarly, the paragraph about B. subtilis-168 doesn't belong to the results section. Same for several other paragraphs in the section. It is highly recommended that the results section focus solely on the experiments conducted for this paper. Other data and results relevant to this work should be dealt with in the discussion section. 

3. In the B. megaterium results section - Fig 5A is inappropriately mentioned as 5B while describing the LB + Sucrose experiments. In the next paragraph, 5B is mislabelled as 3B. Similarly, in figure legends, Fig 5B is  incorrectly described (interchanged with Fig S1 ?). 

4. In fig6, it is not clear why authors illustrate that polydispersity in E coli is less than that in B megaterium. How was this quantified ? Is it avergage PDI of the values mentioned in the tables ? Please provide statistical significance for this conclusion. The arrows here are in contrast to what is mentioned in text - "B. megaterium expressing recombinant has genes produced a higher quality of the polymer (higher MW and lower PDI) than E. coli strains harboring the same plasmids". 

5. Because it was earlier reported that Lactococcus lactis produces a higher yield of HA than what was shown for B megaterium in this paper, it will be useful to the reader if the authors discuss this. What advantages does the current platform provide for HA production compared to the L lactis ?

6. It is recommended that the title of this manuscript be edited to reflect the bacterial strains studied here instead of saying 'different'. 

Author Response

(The authors gave the same response as above.)
